# Prevalence of Pneumococcal Serotypes in Community-Acquired Pneumonia among Older Adults in Italy: A Multicenter Cohort Study

**DOI:** 10.3390/microorganisms11010070

**Published:** 2022-12-26

**Authors:** Andrea Orsi, Alexander Domnich, Stefano Mosca, Matilde Ogliastro, Laura Sticchi, Rosa Prato, Francesca Fortunato, Domenico Martinelli, Fabio Tramuto, Claudio Costantino, Vincenzo Restivo, Vincenzo Baldo, Tatjana Baldovin, Elizabeth Begier, Christian Theilacker, Eva Agostina Montuori, Rohini Beavon, Bradford Gessner, Giancarlo Icardi

**Affiliations:** 1Dipartimento di Scienze della Salute (DiSSal), University of Genoa, 16132 Genoa, Italy; 2Hygiene Unit, Ospedale Policlinico San Martino IRCCS Genoa, 16132 Genoa, Italy; 3Hygiene Unit, Policlinico Foggia Hospital, Department of Medical and Surgical Sciences, University of Foggia, 71122 Foggia, Italy; 4Dipartimento di Promozione della Salute, Materno Infantile, Medicina Interna e Specialistica d’Eccellenza (PROSAMI) “G. D’Alessandro”—Sezione di Igiene—University of Palermo, 90127 Palermo, Italy; 5Dipartimento di Scienze Cardio-Toraco-Vascolari e Sanità Pubblica, University of Padua, 35100 Padua, Italy; 6Global Vaccines, Pfizer Inc., Collegeville, PA 19426, USA; 7Vaccine Medical Affairs, Pfizer Italy, 00188 Rome, Italy

**Keywords:** community-acquired pneumonia (CAP), *Streptococcus pneumoniae*, pneumococcal pneumonia, pneumococcal conjugate vaccines, older adults

## Abstract

Pneumococcal community-acquired pneumonia (CAP) is a leading cause of mortality. Following the introduction of pneumococcal conjugate vaccines (PCVs) in children, a decrease in the burden of the disease was reported. In parallel, an increase in non-vaccine serotypes was also noted. The objective of this study was to assess the current serotype-specific epidemiology of pneumococci among Italian older adults hospitalized for CAP. A prospective study was conducted between 2017 and 2020 in four Italian regions. Subjects aged ≥65 years hospitalized with confirmed CAP were tested for pneumococci using both pneumococcal urinary antigen and serotype-specific urine antigen tests able to identify all 24 serotypes included in the available vaccines. Of the 1155 CAP cases, 13.1% were positive for pneumococci. The most prevalent serotypes were 3 (2.0%), 8 (1.7%), 22F (0.8 %) and 11A (0.7%). These serotypes are all included in the newly licensed PCV20. The serotypes included in PCV13, PCV15 and PCV20 contributed to 3.3%, 4.4% and 7.5% of the CAP cases, respectively. In the context of a low PCV13 coverage among older adults and a high PCV coverage in children, a substantial proportion of CAP is caused by PCV13 serotypes. Higher valency PCV15 and PCV20 may provide additional benefits for the prevention of CAP in vaccinated older adults.

## 1. Introduction

Community-acquired pneumonia (CAP) is a commonly diagnosed disease with a high clinical and economic burden. Together with young children, older adults are the most affected population group [1]. For instance, in Italian adults, the rate of hospitalization following an episode of CAP has been estimated to be 31.8% [2]. *Streptococcus* (*S.*) *pneumoniae* (pneumococci) remains the leading cause of CAP [3]. It has been estimated that about a quarter of all CAP in adults is attributable to pneumococci, although the true fraction is difficult to determine because of the limited sensitivity of the diagnostic tests for non-bacteremic pneumococcal CAP (reviewed in [4]).

On the basis of their polysaccharide capsule, which is considered to be the main determinant of virulence, pneumococci are classified into more than 100 serotypes and all licensed vaccines are based on the capsule antigen [5]. Vaccination is one of the most important public health interventions able to reduce the burden of pneumococcal disease and is recommended for young children, older adults and individuals with underlying chronic conditions. Until recently, two pneumococcal vaccines were available for adults: a 13-valent pneumococcal conjugate (PCV; serotypes 1, 3, 4, 5, 6A, 6B, 7F, 9V, 14, 18C, 19A, 19F and 23F) and 23-valent polysaccharide (PPV23). Although the former induces a T cell-dependent immune response associated with the induction of B cell memory, the latter does not induce a long-lasting immunological memory [6]. Recently, a few studies [7,8] reported a rise in non-PCV13 serotypes detected in invasive pneumococcal disease (IPD) and CAP cases such as 8, 12F and 22F. To address this increase in serotypes not included in PCV13, the higher valency PCV15 (PCV13 serotypes plus serotypes 22F and 33F) and PCV20 (PCV15 serotypes plus serotypes 8, 10A, 11A, 12F and 15B) were developed and recently licensed in the United States (US) and Europe for the immunization of adults aged ≥18 years [9,10].

Studies on the distribution of pneumococcal serotypes have focused mainly on IPD, especially in young children (reviewed in [8,11]). By contrast, data on the serotype distribution in adults with CAP after the introduction of childhood PCV13 immunization are less common. Although IPD surveillance has been in place in Italy since 1994 [12], data regarding the serotype distribution for non-IPD outcomes are collected ad hoc and only a few studies are available. To date, only one study [13] has investigated the role of *S. pneumoniae* in CAP diagnosed among Italian adults. In this small (*n* = 193 older adults) study conducted between October 2011 and October 2012 in Milan, the most frequently detected serotypes in CAP patients were 35F, 3, 24, 6 and 7A and *S. pneumoniae* was detected in 18% of cases.

In Italy, by 2007, most regions issued recommendations on the use of 7-valent PCV (PCV7) in children and in 2010, PCV7 was replaced by PCV13 with a 2 + 1 schedule [14]. In 2017, the Italian Immunization Plan 2017–2019 recommended pneumococcal vaccinations for individuals aged ≥65 years with a single dose of PCV13 followed by PPV23 [15]. Despite these recommendations, the vaccination uptake remains low in both at-risk and older adults [16].

Here, we aimed to analyze the distribution of pneumococcal serotypes among patients aged ≥65 years hospitalized with radiologically confirmed CAP in four geographically dispersed regions of Italy between 2017 and 2020.

## 2. Materials and Methods

### 2.1. Study Design, Participants and Setting

A prospective cohort study was conducted in eight hospitals located in four different Italian regions—namely, Liguria (macro-area of the north-west), Veneto (macro-area of the north-east), Apulia (macro-area of the south) and Sicily (macro-area of islands)—between 2017 and 2020. The secondary objectives of the study are reported here, including the analysis of the burden of pneumococcal CAP (pCAP) and CAP due to pneumococcal vaccine serotypes (VTs). Adults aged ≥65 years residing in the study catchment areas and admitted to the study hospitals for CAP diagnosed both clinically and radiographically were screened for eligibility. CAP was defined as pneumonia acquired outside the hospital or within 48 h following hospital admission with typical chest radiographical patterns (new onset or progressive segmental or multifocal infiltrates) and at least two of the following symptoms: (i) a cough; (ii) purulent sputum or changes in sputum volume/viscosity; (iii) a body temperature of > 38.0 °C or < 36.1 °C; (iv) typical auscultation and/or percussion findings; (v) leukocytosis (white blood cell count > 10,000/mm^3^ or immature neutrophil count > 15%) or leukopenia (white blood cell count < 4500/mm^3^); (vi) a concentration of C-reactive protein > 3 times higher than the upper normal limit; and (vii) hypoxemia with a partial oxygen pressure < 60 mmHg. The exclusion criteria were: (i) onset of CAP symptoms following a hospital admission for at least 48 h in the preceding two weeks; (ii) a diagnosis of CAP ≤ 30 days prior to the assessment; or (iii) a diagnosis of active pulmonary tuberculosis.

The study was conducted in accordance with the Declaration of Helsinki and was approved by the Ethics Committee of the Liguria Region (Genoa, Italy) (resolution P.R. 356REG2016 of 20 September 2016).

### 2.2. Study Procedures and Assessments

Following the informed consent, information on the principal socio-demographics, medical history and standard-of-care laboratory and imaging investigations related to the CAP diagnosis were collected. As a study procedure, a urine sample (30 mL) was obtained from participants and stored at −70 °C until shipment to the central laboratory of the study. Each urine sample was tested with a pneumococcal urinary antigen (PUAT; BinaxNOW *S. pneumoniae*™, Abbott Diagnostics, Scarborough, ME, US) and Pfizer’s proprietary serotype-specific Luminex-based multiplex urinary antigen detection (SSUAD) assays [17,18].

### 2.3. Definitions

pCAP was defined as CAP with the detection of *S. pneumoniae* by the PUAT and/or SSUAD assays. Considering the high diagnostic performance of the SSUAD assays [17,18], samples negative for 24 vaccine serotypes covered by SSUAD but positive in PUAT were assumed to be positive for non-vaccine serotypes (NVTs). Finally, serotypes 1, 5, 7F and 8 were defined as highly invasive whereas the rest were considered to be less invasive [19].

To investigate the short-term changes in the distribution of the serotypes, the study period was divided into three equal periods; namely, 2017–2018 (from 1 June 2017 to 31 May 2018), 2018–2019 (from 1 June 2018 to 31 May 2019) and 2019–2020 (from 1 June 2019 to 31 May 2020).

### 2.4. Data Analysis

Proportions were reported with their 95% confidence intervals (CIs) calculated using Clopper–Pearson’s exact method and the continuous variables were reported as means ± standard deviations (SDs). Additional serotype coverage of the newly available PCV15 and PCV20 (compared with PCV13) was expressed as mean differences with 95% CIs for the dependent proportions. Fisher’s exact test was used to compare the serotype distributions. The Cochran–Armitage test was used to identify the significant short-term trends in the serotype distributions. All analyses were performed in R Stats Package v. 4.1.0 (R Foundation, Vienna, Austria).

## 3. Results

A total of 1155 older adults with clinically and radiologically confirmed CAP and non-missing PUAT and SSUAD results were included in the analysis. The mean age was 79.4 ± 7.6 years, 59.2% were male and 92.2% had at least one underlying medical condition predisposing to pneumococcal disease. Most cases occurred in the fall and winter months and 50.0% were enrolled during the 2018–2019 period (Table 1).

Pneumococcus was detected in 151 (13.1%; 95% CI: 11.2–15.2%) patients with CAP by PUAT or SSUAD. Of these, 35.8% (*n* = 54) tested positive by both PUAT and SSUAD, 27.8% (*n* = 42) by SSUAD only and 36.4% (55) by PUAT only, with the latter category likely reflecting NVT CAP. Among the 96 patients positive by SSUAD, a total of 97 serotype detections occurred as one patient was identified with two serotypes (7F and 15B). Among all CAP cases, the most frequently detected serotypes were 3 (2.0%; 95% CI: 1.3–3.0%), 8 (1.7%; 95% CI: 1.1–2.7%), 22F (0.8%; 95% CI: 0.4–1.5%), 11A (0.7%; 95% CI: 0.3–1.4%) and 9N (0.6%; 95% CI: 0.2–1.2%). The highly invasive serotypes 1, 5, 7F and 8 accounted for a total of 2.1% (95% CI: 1.3–3.1%) of the CAP cases. The serotypes included in PCV15 and PCV20 were identified in 4.4% (95% CI: 3.3–5.8%) and 7.5% (95% CI: 6.1–9.2%) of the CAP cases, respectively (Table 2).

The serotype distributions by age group were similar. Although no statistically significant between-age differences emerged, subjects in the youngest age group of 65–74 years of age (who were also the most prevalent age class in the total pCAP cohort) showed a relatively high prevalence of cases positive only by PUAT (Table 2).

Across the three years, there was a significant (*p* = 0.016) increase in cases positive only by PUAT (2017–2018: 2.9%; 2018–2019: 4.2%; 2019–2020: 7.1%). By contrast, the prevalence of highly invasive VTs significantly decreased (*p* = 0.008) over the study period (2017–2018: 4.6%; 2018–2019: 1.6%; 2019–2020: 1.2%). No significant change (*p* = 0.21) occurred in the proportion of CAP due to the PCV13 serotypes, although the proportion was the lowest during the last study year (2017–2018: 2.9%; 2018–2019: 4.5%; 2019–2020: 1.5%).

## 4. Discussion

To our knowledge, this is the first study describing the contemporary pneumococcal serotype distribution in a large sample of Italian older adults hospitalized for radiologically confirmed CAP. Using urinary antigen testing, 13% of CAP was due to pneumococcus. The most commonly detected serotypes were 3, 8, 22F, 11A, 9N and 15B. Except for serotype 9N, these serotypes are all included in the recently licensed PCV20. Serotypes included in PCV20 represented 7.5% of the radiologically confirmed CAP cases compared with 4.4% and 3.3% for PCV15 and PCV13, respectively. About one-third of the pCAP cases were only diagnosed by PUAT; these likely represented serotypes not included in any available pneumococcal vaccine. The proportion of pCAP due to NVTs increased over the 3-year study period.

Our study highlighted that the serotype distribution between adults and children and between pCAP and IPD may be different. In our study, the most common serotype was serotype 3; during the same time period, serotype 8 was the most common serotype among Italian older adults with IPD [12]. Conversely, in young Italian children, the most frequent serotypes in IPD were 10A and 24F [12]. In this regard, serotype 3 has been shown to be associated with the clinical presentation of pneumonia and pneumonia complications [20] whereas serotype 8 is associated with an invasive disease [21]. The invasive disease potential for serotypes also differs by age and a few serotypes may be more invasive in older adults than in children [22].

In adults, the proportion of CAP due to *S. pneumoniae* in high-income countries varies, with higher values reported in Spain [23] and the United Kingdom (UK) [24] and lower values in the US [25] and Greece [26]. A prospective cohort study on the etiology of CAP in older adults in four Spanish regions reported that 35.0% of CAP cases were due to *S. pneumoniae* and serotypes 3 and 8 were the most commonly detected [23]. Similarly, a CAP surveillance study conducted in the UK between 2013 and 2018 reported that 36.6% of hospitalized CAP cases among adults aged ≥18 years were associated with pneumococcus; serotypes 3 and 8 were more common. In the UK study, no serotype could be determined in a large proportion of pCAP cases [24]. In the US, pneumococcus was identified in 8.8% of adults ≥ 65 years hospitalized for CAP [25] and a Greek study identified pneumococcus in 12.2% of CAP patients [26]. The results of these two studies [25,26] were more similar to our findings. The reasons for the differences between the studies may reflect the true epidemiologic differences in the countries (e.g., based on PCV coverage in children and adults, the use of the influenza vaccine, differences in the waves of co-infecting viral illnesses, the prevalence of underlying diseases or other factors) or methodologic differences such as the patient enrollment criteria, different criteria for defining CAP or differences in the criteria for hospitalizing patients with CAP.

In line with previous reports [23,24,25,26], PCV13 serotype 3 was the most frequent serotype detected in CAP in our study. The disease caused by serotype 3 is often more severe and may evade antibody-mediated clearance. In vitro studies have suggested that serotype 3 may escape the deposition of capsule-specific antibodies and complement this by shedding the capsule antigen into the environment [27]. Although vaccine effectiveness may be lower against serotype 3, a systematic review and meta-analysis by McLaughlin et al. [28] reported a significant pooled PCV13 effectiveness of 52.5% (95% CI: 6.2–75.9%) against serotype 3-hospitalized CAP among directly vaccinated older persons. The authors hypothesized that despite the substantial direct protection, PCV13 may have had a limited impact on the carriage of this serotype [28].

The main limitation of the present study was that the culture-based microbiological investigations were not recorded in the study database and a limited degree of pneumococcal disease misclassification might have occurred. The isolation of *S. pneumoniae* from normally sterile sites is considered to be the gold standard for the diagnosis of bacteremic pCAP. Given the high specificity of SSUAD compared to this gold standard [17,18], false-positive detections for the 24 serotypes included in the SSUAD assay were unlikely. In contrast, the culture of lower respiratory tract secretions has been associated with an acceptable specificity (93–97%) but a limited sensitivity (57–82%), even if the sample is of a high quality [29]. A further limitation of the PUAT is its inability to identify specific serotypes. Furthermore, it has been suggested [30] that the sensitivity of the PUAT for pneumococcal pneumonia is highly variable (from 33.3% to 100%) according to single serotypes. One of the most plausible reasons for this variability is the serotype-specific composition of the C polysaccharide (teichoic acid), which is the PUAT target [30]. In the present study, all specimens were tested in parallel (as opposed to testing in series) using both PUAT and the highly sensitive SSUAD. The parallel testing further increased the overall combined sensitivity [31]. Although the exact distribution of the serotypes among the group positive by PUAT only remains unknown, many (or most) were likely to be NVTs not included in the SSUAD assays. Moreover, the SSUAD assays were validated using samples from patients with bacteremic CAP and its sensitivity for non-bacteremic CAP remains unknown; vaccine-probe studies have suggested that the test likely underestimates the true proportion of PCV-preventable CAP [17,18]. Another shortcoming of our study was the relatively short study period, which limited our ability to evaluate the long-term trends in the serotype evolution following the introduction of PCV13 in Italy. Lastly, the COVID-19 pandemic during the last study year may have impacted the generalizability of the study results, although 94% of the subjects were enrolled before February 2020.

In conclusion, a significant proportion of CAP in older adults in Italy is caused by *S. pneumoniae* and a substantial proportion of this remaining burden is due to serotypes included in the recently licensed PCV20, with serotype 3 and 8 being the most notable serotypes. Our results support the potential of the newly registered PCV15 and PCV20 to broaden PCV coverage in adults as they contain serotypes that are highly prevalent in adult pCAP in Italy and not included in PCV13. For example, serotype 22F is included in both PCV15 and PCV20, and PCV20 also contains serotypes 8 and 11A. Continued efforts are needed to improve pneumococcal vaccine formulations, as indicated by a relatively high proportion of pCAP that was not associated with the serotypes included in currently available PCV formulations.

## Figures and Tables

**Table 1 microorganisms-11-00070-t001:** Characteristics of the study participants.

Characteristic		All-Cause CAP (*N* = 1155), *n* (%)	pCAP (*N* = 151), *n* (%)	pCAP to All-Cause CAP, %
Sex	Female	471 (40.8)	69 (45.7)	14.6
	Male	684 (59.2)	82 (54.3)	12.0
Age, years	65–74	332 (28.7)	50 (33.2)	15.1
	75–79	502 (43.5)	61 (40.3)	12.2
	≥85	321 (27.8)	40 (26.5)	12.5
	Mean (SD)	79.4 (7.6)	79.0 (7.9)	–
Region	Liguria	410 (35.5)	61 (40.3)	14.9
	Veneto	211 (18.3)	33 (21.9)	15.6
	Apulia	193 (16.7)	17 (11.3)	8.8
	Sicily	341 (29.5)	40 (26.5)	11.7
Year	2017–2018	239 (20.7)	30 (19.9)	12.6
	2018–2019	577 (50.0)	81 (53.6)	14.0
	2019–2020	339 (29.3)	40 (26.5)	11.8
Underlying chronic conditions	None	90 (7.8)	16 (10.6)	17.8
	≥1	1065 (92.2)	135 (89.4)	12.7

CAP: community-acquired pneumonia; pCAP: pneumococcal community-acquired pneumonia; SD: standard deviation.

**Table 2 microorganisms-11-00070-t002:** Percent of all community-acquired pneumonia cases due to individual serotypes and grouped by pneumococcal conjugate vaccine (PCV) composition.

Serotype	Age Class (Years), n (%)			
	≥65 (*N* = 1155)	65–74 (*N* = 332)	75–84 (*N* = 502)	≥85 (*N* = 321)
PCV13 serotypes ^1^	38 (3.3)	8 (2.4)	18 (3.6)	12 (3.7)
3	23 (2.0)	5 (1.5)	11 (2.2)	7 (2.2)
19A	3 (0.3)	0 (0)	1 (0.2)	2 (0.6)
5	2 (0.2)	1 (0.3)	1 (0.2)	0 (0)
6A	2 (0.2)	0 (0)	1 (0.2)	1 (0.3)
7F	2 (0.2)	1 (0.3)	1 (0.2)	0 (0)
18C	2 (0.2)	0 (0)	1 (0.2)	1 (0.3)
23F	2 (0.2)	0 (0)	2 (0.4)	0 (0)
4	1 (0.1)	0 (0)	0 (0)	1 (0.3)
19F	1 (0.1)	1 (0.3)	0 (0)	0 (0)
PCV15 serotypes ^2^	51 (4.4)	10 (3.0)	27 (5.4)	14 (4.4)
Additional PCV15 serotypes ^3^	13 (1.1)	2 (0.6)	9 (1.8)	2 (0.6)
22F	9 (0.8)	2 (0.6)	7 (1.4)	0 (0)
33F	4 (0.3)	0 (0)	2 (0.4)	2 (0.6)
PCV20 serotypes ^4^	87 (7.5)	26 (7.8)	40 (8.0)	21 (6.5)
Additional PCV20 serotypes ^5^	36 (3.1)	16 (4.8)	13 (2.6)	7 (2.2)
8	20 (1.7)	8 (2.4)	8 (1.6)	4 (1.2)
11A	8 (0.7)	5 (1.5)	2 (0.4)	1 (0.3)
15B	5 (0.4)	2 (0.6)	1 (0.2)	2 (0.6)
10A	2 (0.2)	0 (0)	2 (0.4)	0 (0)
12F	1 (0.1)	1 (0.3)	0 (0)	0 (0)
Non-PCV serotypes ^6^	10 (0.9)	3 (0.9)	4 (0.8)	3 (0.9)
9N	7 (0.6)	3 (0.9)	2 (0.4)	2 (0.6)
20	2 (0.2)	0 (0)	2 (0.4)	0 (0)
17F	1 (0.1)	0 (0)	0 (0)	1 (0.3)
Non-vaccine type ^7^	55 (4.8)	21 (6.3)	18 (3.6)	16 (5.0)

^1^ Includes serotypes 1, 3, 4, 5, 6A, 6B, 7F, 9V, 14, 18C, 19A, 19F, 23F and the cross-reactive serotype 6C; ^2^ includes serotypes 1, 3, 4, 5, 6A, 6B, 7F, 9V, 14, 18C, 19A, 19F, 22F, 23F, 33F and the cross-reactive serotype 6C; ^3^ includes serotypes 22F and 33F; ^4^ includes serotypes 1, 3, 4, 5, 6A, 6B, 7F, 8, 9V, 10A, 11A, 12F, 14, 15B, 18C, 19A, 19F, 22F, 23F, 33F and the cross-reactive serotypes 6C and 15C; ^5^ includes serotypes 8, 10A, 11A, 12F, 15B and the cross-reactive serotype 15C; ^6^ includes serotypes 2, 9N, 17F and 20; ^7^ samples tested positive in the pneumococcal urinary antigen test but negative in the serotype-specific Luminex-based multiplex urinary antigen detection to all 24 vaccine serotypes. PCV: pneumococcal conjugate vaccine.

## Data Availability

All relevant data are within the manuscript.

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
