# Peer review of "Prevalence of Pneumococcal Serotypes in Community-Acquired Pneumonia among Older Adults in Italy: A Multicenter Cohort Study"

_microorganisms, 2022, doi:10.3390/microorganisms11010070_

Round 1

Reviewer 1 Report

I consider that it is an interesting article, which contributes to the knowledge of the epidemiology of pneumococcus in adults, it is an input that justifies the introduction of pneumococcal vaccines in this group. Congratulations to the authors

Page 2 line78, page 3 line 119:  S. pneumoniae.

Author Response

Comment: I consider that it is an interesting article, which contributes to the knowledge of the epidemiology of pneumococcus in adults, it is an input that justifies the introduction of pneumococcal vaccines in this group. Congratulations to the authors

Reply: Thank you for your interest in our paper. Your comment has been addressed.

Comment: Page 2 line78, page 3 line 119:  S. pneumoniae.

Reply: This has been corrected.

Reviewer 2 Report

Comments to the Author

This cohort study describes serotype prevalence in CAP among older adults in Italy during the 4-year study period. The manuscript also discusses the newly licensed PCV20. This paper is informative and reflects the epidemiology for S. pneumoniae.

Author Response

Comment: This cohort study describes serotype prevalence in CAP among older adults in Italy during the 4-year study period. The manuscript also discusses the newly licensed PCV20. This paper is informative and reflects the epidemiology for S. pneumoniae.

Reply: Thank you for your interest in our paper. No changes required.

Reviewer 3 Report

Title: Prevalence of Pneumococcal Serotypes in Community-Acquired Pneumonia Among Older Adults in Italy: A Multicenter  Cohort Study

The study has a sound methodology.

The analysis has been described well.

The discussion is clear and balanced.

I have a few minor comments:

Line 78         Italicise Genus and species name throughout the manuscript.

Table 2                        How did the authors define the 'Non-vaccine type' for the Samples tested by pneumococcal urinary antigen test? For the 'Non-vaccine type'  category, were the urine samples collected from patients with invasive disease?

Author Response

Comment: The study has a sound methodology. The analysis has been described well. The discussion is clear and balanced. I have a few minor comments.

Reply: Thank you for your interest in our paper. All your comments have been addressed.

Comment: Italicize Genus and species name throughout the manuscript.

Reply: This has been now corrected across the whole manuscript.

Comment: How did the authors define the 'Non-vaccine type' for the Samples tested by pneumococcal urinary antigen test? For the 'Non-vaccine type' category, were the urine samples collected from patients with invasive disease?

Reply: As clearly stated in the Methods, non-vaccine type specimens were those samples that tested positive in the pneumococcal urinary antigen test, but negative in the serotype-specific Luminex-based multi-plex urinary antigen detection to all 24 vaccine serotypes. This has been now specified better in the footnotes of Table 2.

Reviewer 4 Report

General Comments:

After a careful and critical reading of this research study, it was easy to understand that the authors (Orsi, et al.) are reporting for older adults in Italy, before the COVID-19 pandemics, a significant prevalence of Community-Acquired Pneumonia (CAP) caused mainly by serotypes 3 and 8 of Streptococcus pneumoniae. This research study provides important findings to be shared with the scientific community working on the field of medicine and epidemiology of pneumococcal colonization and diseases caused by S. pneumoniae.

By using two urinary antigen tests, the authors detected 151 positive samples for pneumococci among 1,155 urine samples collected from patients with CAP in Italy. Then, the manuscript provides and describes important data and analysis, which were helpful for a good development of the discussion, including the limitations of the study.

However, since the methodology is limited to a non-gold standard and indirect test, which use is not generalized for the purpose of pneumococcal detection, when possible, the authors should consider to include and shortly discuss in the manuscript (Introduction and/or Discussion sections) info about how the Urinary Antigen Tests work, why they cannot be used in urine samples collected from children and why they are restricted to detect a limited number of pneumococcal serotypes.

The answer to these questions will help the readers to understand the importance of the obtained results and findings, strengthening the quality of the manuscript. For example: since serotype 3 is a mucoid serotype, it is more easily detected by this kind of urine indirect assays based on an immuno-chromatographic detection of the pneumococcal C-polysaccharide antigen (Teichoic Acid) of the pneumococcal cell wall.

Minor Comments:

1.   Page 2, Line 78: “…S. pneumoniae…   Convert to italics.

2.   Page 1, Line 35 vs Page 3, Line 144: the number 1155 is written with a different style (1,155). Please check style and correct all throughout the text.

3.   Suggested reference (DOI) to be included: https://doi.org/10.1128/JCM.00787-17

Author Response

Comment: This General Comments. After a careful and critical reading of this research study, it was easy to understand that the authors (Orsi, et al.) are reporting for older adults in Italy, before the COVID-19 pandemics, a significant prevalence of Community-Acquired Pneumonia (CAP) caused mainly by serotypes 3 and 8 of Streptococcus pneumoniae. This research study provides important findings to be shared with the scientific community working on the field of medicine and epidemiology of pneumococcal colonization and diseases caused by S. pneumoniae. By using two urinary antigen tests, the authors detected 151 positive samples for pneumococci among 1,155 urine samples collected from patients with CAP in Italy. Then, the manuscript provides and describes important data and analysis, which were helpful for a good development of the discussion, including the limitations of the study.

Reply: Thank you for your interest in our paper. All your comments have been addressed.

Comment: However, since the methodology is limited to a non-gold standard and indirect test, which use is not generalized for the purpose of pneumococcal detection, when possible, the authors should consider to include and shortly discuss in the manuscript (Introduction and/or Discussion sections) info about how the Urinary Antigen Tests work, why they cannot be used in urine samples collected from children and why they are restricted to detect a limited number of pneumococcal serotypes. The answer to these questions will help the readers to understand the importance of the obtained results and findings, strengthening the quality of the manuscript. For example: since serotype 3 is a mucoid serotype, it is more easily detected by this kind of urine indirect assays based on an immuno-chromatographic detection of the pneumococcal C-polysaccharide antigen (Teichoic Acid) of the pneumococcal cell wall.

Reply: Thank you for this comment. As suggested, we have now discussed the issue of high variability of the PUAT in terms of its serotype-specific sensitivity, which is likely driven by the composition of the teichoic acid. The suggested paper by Shoji et al. (see also your minor comment below) has been cited. Indeed, one of the main study strengths is that all specimens were tested in parallel (as opposed to the testing in series), using both PUAT and highly sensitive SSUAD. The parallel testing increased further the overall combined sensitivity. The text has been amended accordingly.

Comment: Minor Comments: 1.   Page 2, Line 78: “…S. pneumoniae…   Convert to italics.

Reply: This has been corrected.

Comment: 2.   Page 1, Line 35 vs Page 3, Line 144: the number 1155 is written with a different style (1,155). Please check style and correct all throughout the text.

Reply: The whole manuscript has been revised and the style has been amended. 

Comment: 3.   Suggested reference (DOI) to be included: https://doi.org/10.1128/JCM.00787-17

Reply: This reference has been added to the manuscript (as per your previous comment).